# Blood-based epigenome-wide analyses of 19 common disease states: A longitudinal, population-based linked cohort study of 18,413 Scottish individuals

Robert F. Hillary[1], Daniel L. McCartney[1], Hannah M. Smith[1], Elena Bernabeu[1], Danni A. Gadd[1], Aleksandra D. Chybowska[1], Yipeng Cheng[1], Lee Murphy[2], Nicola Wrobel[2], Archie Campbell[1], Rosie M. Walker[1,3], Caroline Hayward[1,4], Kathryn L. Evans[1], Andrew M. McIntosh[1,5], Riccardo E. Marioni[1]*

1 Centre for Genomic and Experimental Medicine, Institute of Genetics and Cancer, University of Edinburgh, Edinburgh, United Kingdom, 2 Edinburgh Clinical Research Facility, University of Edinburgh, Edinburgh, United Kingdom, 3 School of Psychology, University of Exeter, Exeter, United Kingdom, 4 Medical Research Council Human Genetics Unit, Institute of Genetics and Cancer, University of Edinburgh, Edinburgh, United Kingdom, 5 Division of Psychiatry, University of Edinburgh, Royal Edinburgh Hospital, Edinburgh, United Kingdom

* riccardo.marioni@ed.ac.uk

**Data Availability Statement:** EWAS summary statistics are available on the EWAS Catalog (http://ewascatalog.org/?query=10.1101/2023.01.10.23284387). According to the terms of consent for Generation Scotland participants, access to data must be reviewed by the Generation Scotland

## Abstract

### Background

DNA methylation is a dynamic epigenetic mechanism that occurs at cytosine-phosphate-guanine dinucleotide (CpG) sites. Epigenome-wide association studies (EWAS) investigate the strength of association between methylation at individual CpG sites and health outcomes. Although blood methylation may act as a peripheral marker of common disease states, previous EWAS have typically focused only on individual conditions and have had limited power to discover disease-associated loci. This study examined the association of blood DNA methylation with the prevalence of 14 disease states and the incidence of 19 disease states in a single population of over 18,000 Scottish individuals.

### Methods and findings

DNA methylation was assayed at 752,722 CpG sites in whole-blood samples from 18,413 volunteers in the family-structured, population-based cohort study Generation Scotland (age range 18 to 99 years). EWAS tested for cross-sectional associations between baseline CpG methylation and 14 prevalent disease states, and for longitudinal associations between baseline CpG methylation and 19 incident disease states. Prevalent cases were self-reported on health questionnaires at the baseline. Incident cases were identified using linkage to Scottish primary (Read 2) and secondary (ICD-10) care records, and the censoring date was set to October 2020. The mean time-to-diagnosis ranged from 5.0 years (for chronic pain) to 11.7 years (for Coronavirus Disease 2019 (COVID-19) hospitalisation). The 19 disease states considered in this study were selected if they were present on the World

Access Committee. Applications should be made to access@generationscotland.org.

**Funding:** This research was funded in whole, or in part, by the Wellcome Trust (104036/Z/14/Z, 216767/Z/19/Z, 220857/Z/20/Z to AMM; 108890/Z/15/Z to DAG and 218493/Z/19/Z to HMS). This work was supported by the British Heart Foundation (Immediate Fellowship FS/IPBSRF/22/27042 to RFH), Alzheimer's Society (AS-PG-19b-010 to REM and supports EB), and the Medical Research Council (U. MC_UU_00007/10 to CH). The Generation Scotland study was also awarded funding and supported by the Chief Scientist Office of the Scottish Government Health Directorates (CZD/16/6), the Scottish Funding Council (HR03006) and Wellcome (104036/Z/14/Z to AMM). AMM acknowledges further support by Wellcome (216767/Z/19/Z and 220857/Z/20/Z), United Kingdom Research and Innovation Medical Research Council (MC_PC_17209, MR/W014386/1 and MR/S035818/1) and the European Union H2020 (SEP-210574971). HMS and DAG are supported by Wellcome through the Translational Neuroscience PhD Programme (218493/Z/19/Z to HMS and 108890/Z/15/Z to DAG). YC is supported by the University of Edinburgh and University of Helsinki joint PhD program in Human Genomics. ADC is supported by a Medical Research Council PhD Studentship in Precision Medicine with funding by the Medical Research Council Doctoral Training Programme and the University of Edinburgh College of Medicine and Veterinary Medicine. RFH receives salary support from the British Heart Foundation (FS/IPBSRF/22/27042), EB receives salary support through Alzheimer's Society (AS-PG-19b-010). The funders had no role in study design, data collection and analysis, decision to publish, or preparation of the manuscript. o Wellcome: https://wellcome.org/ o British Heart Foundation: https://www.bhf.org.uk/ o Alzheimer's Society: https://www.alzheimers.org.uk/ o Medical Research Council: https://www.ukri.org/councils/mrc/ o Chief Scientist Office: https://www.cso.scot.nhs.uk/ o Scottish Funding Council: https://www.sfc.ac.uk/ o UK Research and Innovation: https://www.ukri.org/ o European Union H2020: https://research-and-innovation.ec.europa.eu/funding/funding-opportunities/funding-programmes-and-open-calls/horizon-2020_en.

**Competing interests:** I have read the journal's policy and the authors of this manuscript have the following competing interests: RFH has received speaker fees from Illumina and acts as a scientific consultant to Optima Partners. DAG acts as a scientific consultant for Optima Partners. LM has received speaker and consultancy fees from

Health Organisation's 10 leading causes of death and disease burden or included in baseline self-report questionnaires. EWAS models were adjusted for age at methylation typing, sex, estimated white blood cell composition, population structure, and 5 common lifestyle risk factors. A structured literature review was also conducted to identify existing EWAS for all 19 disease states tested. The MEDLINE, Embase, Web of Science, and preprint servers were searched to retrieve relevant articles indexed as of March 27, 2023. Fifty-four of approximately 2,000 indexed articles met our inclusion criteria: assayed blood-based DNA methylation, had >20 individuals in each comparison group, and examined one of the 19 conditions considered. First, we assessed whether the associations identified in our study were reported in previous studies. We identified 69 associations between CpGs and the prevalence of 4 conditions, of which 58 were newly described. The conditions were breast cancer, chronic kidney disease, ischemic heart disease, and type 2 diabetes mellitus. We also uncovered 64 CpGs that associated with the incidence of 2 disease states (COPD and type 2 diabetes), of which 56 were not reported in the surveyed literature. Second, we assessed replication across existing studies, which was defined as the reporting of at least 1 common site in >2 studies that examined the same condition. Only 6/19 disease states had evidence of such replication. The limitations of this study include the nonconsideration of medication data and a potential lack of generalizability to individuals that are not of Scottish and European ancestry.

## Conclusions

We discovered over 100 associations between blood methylation sites and common disease states, independently of major confounding risk factors, and a need for greater standardisation among EWAS on human disease.

## Author summary

### Why was this study done?

- Blood DNA methylation can inform us about the biological mechanisms that underlie common disease states. Epigenome-wide association studies (EWAS) investigate whether the proportion of methylation at loci termed CpG sites (cytosine-phosphate-guanine dinucleotides) associate with health outcomes of interest.

- There is a need for large-scale EWAS that probe for epigenetic signals across a wide range of conditions as well as a structured literature review to inform the utility of this approach in identifying disease-relevant loci.

### What did the researchers do and find?

- DNA methylation was assayed at 752,722 CpG sites using whole-blood samples from 18,413 volunteers, which were collected at the study baseline of Generation Scotland (2006 to 2011).

Illumina. AMM has received research support from Eli Lilly, Janssen, and the Sackler Foundation. AMM has also received speaker fees from Illumina and Janssen and consulting fees. REM has received a speaker fee from Illumina and is an advisor to the Epigenetic Clock Development Foundation and Optima Partners.

**Abbreviations:** AD, Alzheimer's dementia; CKD, chronic kidney disease; COPD, chronic obstructive pulmonary disease; COVID-19, Coronavirus Disease 2019; CpG, cytosine-phosphate-guanine dinucleotide; DALY, disability-adjusted life year; DNAm, DNA methylation; eGFR, estimated glomerular filtration rate; EWAS, epigenome-wide association studies; GO, Gene Ontology; GS, Generation Scotland; GWAS, genome-wide association studies; IBD, inflammatory bowel disease; KEGG, Kyoto Encyclopaedia of Genes and Genomes; OSCA, OmicS-data-based Complex trait Analysis; SD, standard deviation; TPRG1, Tumour protein P63 Regulated 1; UBIAD1, UbiA Prenyltransferase Domain Containing 1; WBC, white blood cell.

- EWAS tested for associations between differential methylation at CpG sites and the prevalence and incidence of 14 and 19 disease states, respectively. Prevalence and incidence data were derived from self-report questionnaires and electronic health record linkage, respectively.

- We identified over 100 CpG associations with 4 prevalent conditions (breast cancer, chronic kidney disease, ischemic heart disease, and type 2 diabetes) and 2 incident conditions (chronic obstructive pulmonary disease and type 2 diabetes). We also found poor replicability among existing studies with lung cancer showing the highest degree of replication (17% of sites replicated in at least 2 studies).

### What do these findings mean?

- Blood DNA methylation could act as a peripheral marker of several common disease states including breast cancer, cardiopulmonary disease, and type 2 diabetes.

- As population biobank resources expand, studies that examine the same condition should reach consensus on covariate strategies, phenotype definitions, and reporting guidelines.

## 1. Introduction

Epigenetic modifications to DNA represent an important mechanism by which the environment interacts with the genome [1]. DNA methylation (DNAm) is one of the best-studied epigenetic mechanisms and involves the addition of chemical tags termed methyl groups to DNA, typically in the context of cytosine-phosphate-guanine dinucleotides (CpG sites). Factors such as diet, stress, and smoking behaviours may influence the process of methylation. The addition of these chemical tags can alter whether, and to what extent, a gene is active. In contrast to genetic sequence variation, these modifications are reversible and can modulate gene expression in cell- and tissue-specific manners [2]. Genome-wide patterns of DNAm are most commonly assayed using microarray-based technologies such as the Illumina HumanMethylation 450K and HumanMethylationEPIC arrays. The arrays permit a cost-effective assessment of DNAm at a scale required for large-scale population health studies [3,4].

Epigenome-wide association studies (EWAS) examine associations between the proportion of methylation at CpG sites and health outcomes of interest, such as chronic disease states [5]. Primarily, EWAS have been conducted using whole-blood DNAm. Patterns of DNAm identified in blood do not necessarily mirror DNAm patterns in distal or disease-relevant tissues such as nervous tissue for Alzheimer's disease [6,7]. However, blood sampling represents a minimally invasive route for scalable biomarker measurement. Blood-based EWAS have also implicated differential methylation at individual loci as candidate markers of disease risk. For example, *TXNIP* and *ABCG1* are important regulators of glucose and cholesterol metabolism, respectively. Hypomethylation within *TXNIP* (cg19693031) and *ABCG1* hypermethylation (cg06500161) have been associated with type 2 diabetes risk across individuals of multiple ancestries [8–11].

Existing EWAS on common diseases can be broadly categorised into prevalence analyses (i.e., cross-sectional) and incidence analyses (i.e., longitudinal assessment of incident cases in unaffected individuals). EWAS have often relied on modest sample sizes (<1,000 individuals),

which has limited the discovery of loci that associate with disease states. Meta-analyses can increase power but may be vulnerable to between-study heterogeneities. There is a need for large-scale EWAS that examine the prevalence and incidence of multiple disease states in a single population. These analyses would help to establish the relevance of blood methylation as a peripheral marker of common disease states. Furthermore, there is a need for structured literature reviews to assess the level of agreement in locus discovery among existing EWAS that examine the same condition. A synthesis of the level of concordance between published association studies would aid in evaluating the utility of epigenome-wide analyses as an avenue for identifying risk mechanisms underlying common disease states.

Here, we utilise Generation Scotland: the Scottish Family Health Study (GS), a large cohort with DNAm data ($n$ = 18,413). We hypothesise that differential methylation at CpG sites associates with the prevalence of 14 conditions and the incidence of 19 disease states. First, we integrate blood DNAm and self-reported disease data from questionnaires answered at the study baseline to perform EWAS on 14 prevalent disease states (cross-sectional analyses). Second, we conduct EWAS on 19 incident disease states ascertained through electronic health record linkage over up to 14 years of follow-up (longitudinal analyses). Third, we perform a structured literature review to identify blood-based EWAS findings on all 19 disease states considered in this study. We examine whether findings in this study replicate previous analyses and quantify the level of agreement within previously published studies. Fourth, we employ genetic colocalisation analyses to determine whether DNAm levels at the loci identified in our EWAS and disease risk mechanisms are likely influenced by shared or distinct genetic variants. These analyses would help to determine whether DNAm is an important molecular mechanism connecting genetic risk to disease endpoints. **Fig 1** provides a visual summary of the study design.

## 2. Methods

### 2.1. Ethics statement

All components of Generation Scotland received ethical approval from the NHS Tayside Committee on Medical Research Ethics (REC Reference Number: 05/S1401/89). Generation Scotland has also been granted Research Tissue Bank status by the East of Scotland Research Ethics Service (REC Reference Number: 20-ES-0021), providing generic ethical approval for a wide range of uses within medical research. Written informed consent was obtained from all participants. This study was performed in accordance with the Helsinki declaration.

### 2.2. Generation Scotland cohort

Generation Scotland, or GS, is a large family-structured cohort study that consists of 24,000 individuals from across Scotland. Participants were identified via Community Health Index numbers, with the support of Scottish Practices and Professionals Involved in Research. The initial phase of recruitment (2006 to 2010) focussed on the Glasgow and Tayside regions of Scotland and was later extended to Ayrshire, Arran, and the Northeast of Scotland. Individuals must have been aged between 35 and 65 years, had ≥1 first-degree relative and ≥1 full sibling. The age range was later broadened to 18 to 65 years. Family members of probands were also invited to partake in the study. In total, 23,960 individuals were recruited, which encompassed 6,665 probands, 16,007 family members, and 1,288 individuals who self-volunteered without invitation. There were 5,573 families with a mean size of 4 members and 1,400 participants without relatives. The median age at baseline was 47 years and the sample was 59% female [12,13]. Detailed health and lifestyle information were collected via questionnaires at the study baseline alongside venepuncture to obtain whole blood samples from which DNAm was

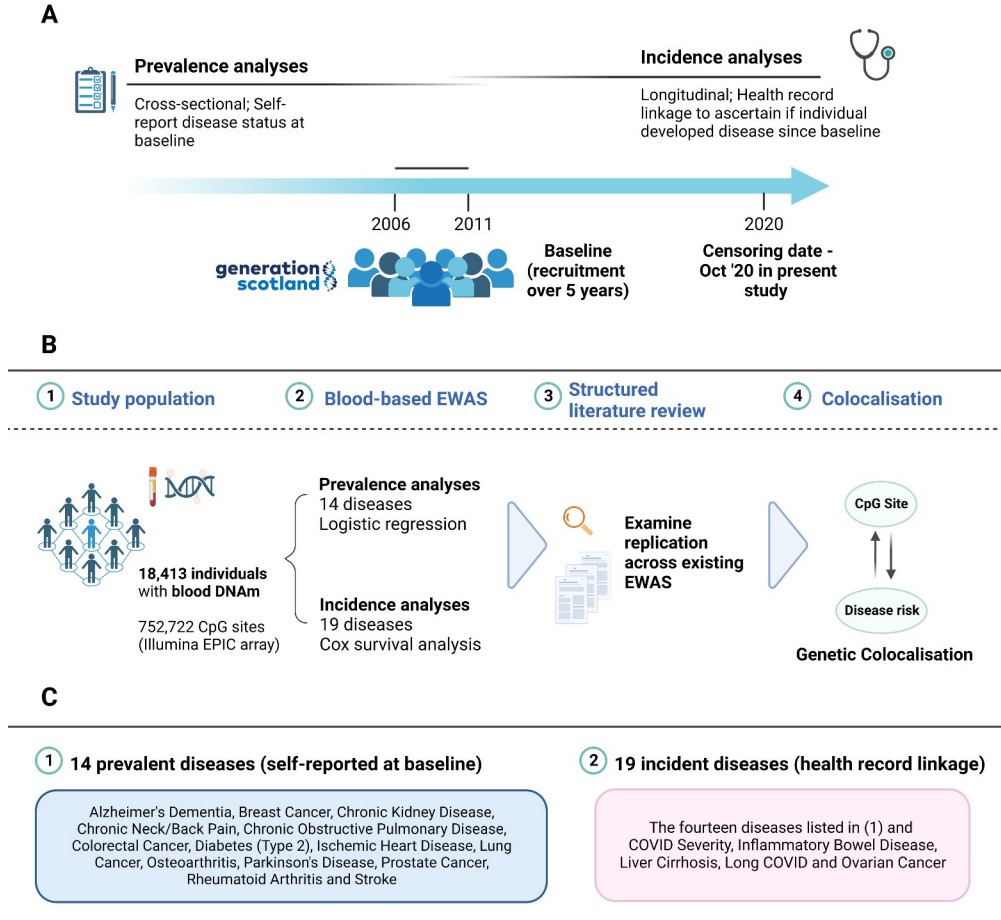

**Fig 1. Study design for epigenome-wide analyses on prevalent and incident disease states in Generation Scotland.** (A) Recruitment for Generation Scotland took place between 2006 and 2011. Prevalence analyses: participants self-reported disease status and donated blood samples at the study baseline. Incidence analyses: linked healthcare data were used to determine if participants who were free from a particular condition at baseline went on to develop the condition over up to 14 years of follow-up. Controls were free of the disease at the baseline and during follow-up. (B). Blood DNAm at baseline was available for 18,413 participants. The mean age was 47.5 years and the sample was 58.8% female. EWAS tested for associations between blood CpG methylation and the prevalence of 14 disease states at baseline or the incidence (time-to-onset) of 19 disease states during follow-up. The mean time-to-diagnosis ranged from 5.0 years (for chronic pain) to 11.7 years (for COVID-19 hospitalisation). Significant findings were tested for replication in existing studies via a structured literature review. Replication within existing studies was also investigated. Colocalisation analyses were employed to help dissect whether associations between DNAm and disease states reflected shared or distinct genetic architectures. (C). The first box lists the 14 self-reported disease states at the study baseline, which were included in this study. The second box lists the 19 incident disease states identified through electronic health record linkage. They include the same 14 conditions listed in the first box along with 5 additional disease states. Of note, prevalent AD reflected family history of the disease due to the young mean age of the sample at baseline, whereas incident AD reflected diagnosed disease. Image created using Biorender.com. AD, Alzheimer's dementia; COVID-19, Coronavirus Disease 2019; CpG, cytosine-phosphate-guanine dinucleotide; DNAm, DNA methylation; EWAS, epigenome-wide association studies.

assayed. This study is reported as per the Strengthening the Reporting of Observational Studies in Epidemiology (STROBE) guideline (S1 STROBE Checklist).

The present study does not have a registered prospective protocol. An unpublished, informal analysis plan was made and discussed among study authors prior to the implementation of statistical analyses (August 2022). There were no significant changes to the analysis plan following informal review among the study authors, with the exception of pathway enrichment and outlier sensitivity analyses following peer review.

## 2.3. Preparation of DNA methylation data

Whole-blood DNAm was measured using the Illumina Infinium MethylationEPIC array. DNAm profiling of the GS samples was carried out by the Genetics Core Laboratory at the Edinburgh Clinical Research Facility, Edinburgh, Scotland. Methylation typing was performed in 3 distinct sets. Quality control steps are detailed in full in **S1 Text**. Following quality control, there were 5,087, 4,450, and 8,876 individuals within Sets 1, 2, and 3, respectively. Set 1 contained related individuals. Set 2 consisted of individuals who were unrelated to each other and those in Set 1. Set 3 consisted of related individuals, and individuals related to those in Sets 1 and 2. The sets were combined and dasen normalisation was performed across all individuals [14]. Linear regression models were used to adjust methylation M-values for chronological age, sex, and experimental batch (factor with 121 levels, i.e., individuals were assayed across 121 unique batches). Residualised M-values were taken forward for analyses. In total, 752,722 probes and 18,413 individuals passed quality control criteria and were considered as a single analytical sample in our analyses.

## 2.4. Preparation of disease phenotypes

Nineteen common disease states were considered across prevalence and incidence analyses: (i) Alzheimer's dementia (AD); (ii) breast cancer; (iii) chronic kidney disease (CKD); (iv) chronic neck and/or back pain; (v) chronic obstructive pulmonary disease (COPD); (vi) colorectal cancer; (vii) Coronavirus Disease 2019 (COVID-19) severity (requiring hospitalisation); (viii) inflammatory bowel disease (IBD); (ix) ischemic heart disease; (x) liver cirrhosis; (xi) long COVID; (xii) lung cancer; (xiii) osteoarthritis; (xiv) ovarian cancer; (xv) Parkinson's disease; (xvi) prostate cancer; (xvii) rheumatoid arthritis; (xviii) stroke; and (xix) type 2 diabetes. Outcomes were selected if they were present among the 10 leading causes of death in high-income countries, the 10 leading causes of disease burden (disease-adjusted life years (DALYs)) in high-income countries or self-reported conditions at the baseline [15–17]. Depression was not considered as it is included in an ongoing meta-analysis EWAS. Although asthma can occur at any age, it has a higher prevalence among children aged 0 to 17 years than in adults. It was therefore excluded from the present analyses that used an adult sample with a broad age profile [18].

Self-report data were used for 12 disease states in cross-sectional analyses of disease prevalence. Self-reported parental history of AD was used a proxy variable for AD. Analyses on self-reported parental history of AD were restricted to participants who were >45 years at baseline. This ensured that only participants whose parents were likely old enough at baseline to be at risk of AD were considered (i.e., >65 years). The CKD Epidemiology Collaboration, or CKD-EPI, equation was implemented to estimate glomerular filtration rate (eGFR) at baseline. Individuals with an eGFR <60 ml/min/1.73 m$^2$ were deemed to have CKD [19]. Therefore, 14 disease phenotypes were considered in prevalent analyses.

All 19 phenotypes were included in longitudinal analyses via linkage to electronic health records (with the exception of self-reported long COVID). The primary and secondary care codes used to define incident phenotypes are available in S1 **Appendix**. Prevalent cases from the study baseline were excluded for these analyses as were those where record linkage provided evidence of a diagnosis prior to baseline. Therefore, incident cases included those diagnosed after baseline who had died and those who received a diagnosis and remained alive. Controls were censored if they were free of a diagnosis at the time of death or at the end of the follow-up period. Further information on the preprocessing of incident phenotypes, including COVID phenotypes, is available in **S2 Text**.

## 2.5. Epigenome-wide association studies on prevalent disease

First, logistic regression models were used to adjust prevalent phenotypes for chronological age and sex, with the exception of breast cancer and prostate cancer, which were adjusted for age after restricting the cohort to females and males, respectively. Second, linear regression models were used for EWAS via the OSCA (OmicS-data-based Complex trait Analysis) software [20]. Residuals from logistic regression models were entered as the dependent variable and age-, sex-, and batch-adjusted CpG M-values represented the independent variable. This strategy was employed to reduce computational burden. A Bonferroni significance threshold was set at $p < 2.6 \times 10^{-9}$ (= $3.6 \times 10^{-8}$/14 phenotypes) [21]. Two models with different covariate strategies were employed, as described below:

1. **Basic model:** Phenotype and CpG M-values, processed as described above, and 5 Houseman-estimated white blood cell (WBC) proportions as fixed effect covariates [22]. Six cell types are estimated from the Houseman method. However, their proportions sum to 100%. Therefore, the percentage of granulocytes was not included in this analysis given that it is collinear with the other 5 cell types. The basic model was as follows:

    Phenotype (residuals) ~ CpG M-values (residuals) + 5 methylation-predicted WBC proportions.

2. **Fully adjusted model:** additional adjustments for 5 common lifestyle factors, which were alcohol consumption, body mass index, deprivation index (Scottish Index of Multiple Deprivation), methylation-based smoking score (EpiSmokEr) [23], and years of education. Body mass index was log transformed prior to analysis. Furthermore, multidimensional scaling was applied to GS genotype data to obtain an estimate of population structure. The first 20 genetic principal components were extracted and included in our analytical models. The fully adjusted model was as follows:

    Phenotype (residuals) ~ CpG M-values (residuals) + 5 methylation-predicted WBC proportions + alcohol consumption (units/week) + log(body mass index (kg/m$^2$)) + deprivation index (Scottish Index of Multiple Deprivation) + education (an 11-category ordinal variable) + methylation-based smoking score (EpiSmokEr) + 20 genetic PCs (population structure).

Results from basic and fully adjusted models are presented within the main text. Both models are included to assess the effects of lifestyle factors on associations between methylation sites and common disease states. Some covariates may be more appropriate for one disease state over another (e.g., body mass index for type 2 diabetes versus cigarette smoking for COPD). However, all 5 risk factors are included in an effort to capture the most common environmental and lifestyle risk factors across a broad range of disparate conditions. We do not further present unadjusted analyses (i.e., using DNAm data that are not adjusted for age, sex, and batch effects) given the strong, possible confounding effects of age and technical variation on associations between CpG methylation and age-related disease states. We also did not initially adjust for family structure in our models. However, we later ran a series of sensitivity analyses (outlined in Section 3.5), including adjustment for relatedness between participants.

## 2.6. Epigenome-wide association studies on incident disease

First, Cox proportional hazards models were used to adjust incident phenotypes for age at baseline and sex (17/19 phenotypes). Only age was included for breast, ovarian, and prostate cancer. Time-to-onset for the disease, or censoring, was the survival outcome in Cox proportional hazards models. Only individuals with an age at event or censoring ≥65 years were

considered for AD. As outlined above, controls were censored at the time of death or at the end of the follow-up period. Logistic regression models were used to adjust 2 remaining COVID phenotypes prior to EWAS analyses. Cox models were not employed for COVID phenotypes owing to the limited differences in time-to-event data between individuals with positive COVID diagnoses. Whereas DNAm was corrected for age at baseline (as well as sex and batch), COVID phenotypes were adjusted for sex and age at COVID testing or diagnosis. Here, age at COVID testing or diagnosis was considered given the variation in time elapsed between baseline visits (between 2006 and 2011) and the onset of the COVID pandemic. Second, martingale residuals or logistic regression residuals were extracted and included as dependent variables in OSCA. A Bonferroni-corrected significance threshold was set at $p < 1.9 \times 10^{-9}$ (= $3.6 \times 10^{-8}$/19 phenotypes). Basic and fully adjusted models were employed, as described in the previous section. Methods for sensitivity EWAS analyses are detailed under **S3 Text**.

### 2.7. Pathway enrichment analyses

Enrichment was assessed among Kyoto Encyclopaedia of Genes and Genomes (KEGG) pathways and Gene Ontology (GO) terms using the gometh() function in the R package *missMethyl* [24]. This function models the relationship between the number of probes per gene and the probability of being selected, accounting for the selection bias associated with probe-dense genes. The top 100 CpGs (i.e., smallest EWAS *p*-values) from each fully adjusted model were included as input features. There were 33 such models for consideration (14 prevalent and 19 incident models). Pathways with an FDR-adjusted *p*-value < 0.05 were deemed significant.

### 2.8. Structured literature review on blood-based EWAS of common disease

MEDLINE, Embase (Ovid interface, 1980 onwards), Web of Science (core collection, Thomson Reuters), and preprint servers were searched to identify relevant articles indexed as of March 27, 2023. The initial search dates were between August 1, 2022 and August 31, 2022, and later updated and performed again on March 27, 2023. We used the following search terms or their synonyms appropriate to each database: ("blood".mp OR "whole blood".mp OR "peripheral blood.mp") AND ("EWAS" OR exp "epigenome-wide*" / OR exp "epigenome-wide association" /) AND (the disease of interest, e.g., "COPD" OR "chronic obstructive pulmonary disease"). The search strategy returned approximately unique 2,000 articles, of which 54 passed inclusion criteria. Inclusion criteria were as follows: (i) original research article; (ii) EWAS performed with blood DNAm; (iii) there were at least 20 individuals in each comparison group (i.e., cases and controls); and (iv) the study examined at least one of the 19 common disease states outlined in our study.

Here, we make an important distinction between systematic reviews and our structured literature review. The structured search of the literature was intended to identify appropriate studies for look-up analyses using a predefined and agreed list of search terms. This is similar to systematic reviews in that search terms are used to systematically screen literature databases. However, our approach differed from a systematic review in that no original or meta-analyses were performed using data from the literature beyond a look-up analysis of CpGs identified in these studies. Unlike a systematic review, the approach also does not provide an estimate for a clinical question and rather summarises the current EWAS literature.

First, we wished to examine whether the CpG associations identified in our study had been previously described. A CpG site was declared as novel in our study if it was not previously reported at experiment-wise significance thresholds deemed by each of the 54 studies. Of note, these studies used different significance thresholds. Several studies did not make their full

summary statistics available, which prohibited the use of a common significance threshold for look-up analyses. However, the studies also differed from one another with respect to methylation arrays, phenotype definitions, and covariate strategies. We focussed on unique CpGs rather than unique genomic locations. Look-up analyses were performed separately for each condition following our structured literature review. Second, we aimed to determine the level of agreement among studies that examined the same condition with respect to locus discovery. Here, our study was ignored as we were only interested in the previous literature for this analysis. A CpG site or its gene (if available) was considered to be replicated if it was reported as significant (at thresholds set by each study) in at least 2 studies that examined the same condition. While focusing on genes alone may neglect intergenic CpGs, the CpG-level and gene-level look-up analyses are included together in an effort to capture as much information as possible from disparate studies in the literature.

### 2.9. Colocalisation analyses

Colocalisation analyses required GWAS summary statistics for CpG sites (i.e., methylation Quantitative Trait Loci–mQTLs, trait 1) and for respective disease states (trait 2; [25–30]). The GoDMC mQTL resource represents the largest mQTL study to date in terms of sample size but only focused on 450k array sites [31]. Therefore, the GoDMC resource was utilised for sites that are common to the EPIC and 450k arrays. However, mQTL analyses were also conducted in GS due to the need to generate mQTL summary statistics for sites present on the EPIC array only (S4 Text). In instances where CpGs had associations in both GS and GoDMC, we used the following criteria to determine which dataset to retain: (i) the dataset must have >10 genetic variants available and (ii) if both datasets satisfy (i), then retain the dataset with the larger sample size. Of note, GS served as the replication cohort within the original GoDMC analyses. Effect sizes in GS and GoDMC showed correlation coefficients of 0.97 and 0.96 for *cis* and *trans* variants, respectively, in the original GoDMC publication [31]. We observed a similar coefficient of 0.97 between effect sizes for the subset of CpGs used in our colocalisation analyses. Therefore, there was likely little heterogeneity between the data sources used in our workflow.

The coloc.abf() function in the R package *coloc* was used to test for colocalisation and default parameters were applied (version 5.1.0) [32]. SNPs ±1 Mb surrounding each CpG site were extracted from mQTL datasets (i.e., GS or GoDMC, trait 1) and disease GWAS summary statistics (trait 2). The method tests for 5 mutually exclusive hypotheses, H0: there are no causal variants for either trait in the tested region; H1 and H2: causal variant for trait 1 and trait 2 only, respectively; H3: distinct causal variants for both traits; and H4: the traits share a causal variant. Posterior probabilities ≥95% for H4 provided strong evidence in favour of colocalisation.

## 3. Results

### 3.1. Demographics and disease counts in Generation Scotland

The mean age of the sample was 47.5 years ($n$ = 18,413, standard deviation (SD) = 14.9) and the sample was 58.8% female. Summary data for demographic variables are presented in **Table 1**. Additional data on covariates and disease counts are displayed in **S1**–**S3 Tables**. The number of self-reported cases for prevalent disease at baseline ranged from 34 participants with Parkinson's disease to 5,296 with chronic neck and/or back pain, respectively (basic model). Further, the number of cases with incident disease since baseline (derived from health record linkage) ranged from 31 for severe COVID (hospitalisation from COVID-19 infection) to 1,886 for chronic neck and/or back pain. Associations between covariates and disease states

**Table 1. Summary of demographic variables in the Generation Scotland cohort.**

| Phenotype | Units | n | Mean | SD |
|---|---|---|---|---|
| Age | years | 18,413 | 47.5 | 14.9 |
| Alcohol Consumption | units/week | 16,705 | 11.0 | 13.0 |
| Body Mass Index | kg/m$^2$ | 18,299 | 27.0 | 5.2 |
| DNAm smoking score (EpiSmokEr) | - | 18,413 | 1.4 | 4.3 |
| | | n | Median | IQR |
| Education | 11-category ordinal variable | 17,389 | 4 | 3 |
| Scottish Index of Multiple Deprivation | rank | 17,287 | 4,331 | 3,115 |
| | | n | n-female | % female |
| Sex | - | 18,413 | 10,833 | 58.8 |

DNAm, DNA methylation; IQR, interquartile range; SD, standard deviation.

Education was measured as an ordinal variable: 0, 0 years; 1, 1–4 years; 2, 5–9 years; 3, 10–11 years; 4, 12–13 years; 5, 14–15 years; 6, 16–17 years; 7, 18–19 years; 8, 20–21 years; 9, 22–23 years; 10, ≥24 years.

are displayed in **S4 and S5 Tables** for prevalent and incident disease states, respectively (also available in **S1 and S2 Figs**).

## 3.2. Epigenome-wide analyses of prevalent disease

We first tested for cross-sectional associations between blood CpG methylation and 14 disease states at the study baseline. There were 1,340 significant associations across 10 diseases in a basic model that adjusted for age, sex, and estimated blood cell proportions ($p < 2.6 \times 10^{-9}$; **Fig 2A, S6 Table**). Over 90% of these associations ($n = 1,246$) were attributed to type 2 diabetes ($n = 703$ associations, 52.5%), COPD ($n = 301$, 22.5%), and chronic pain ($n = 242$, 18.1%). Genomic inflation factors ranged from 0.8 to 1.6 across all basic models (**S7 Table**). Look-up analyses in the EWAS Catalog showed that 617/1,340 associations involve CpGs that were previously associated with common disease risk factors including body mass index, smoking, and alcohol consumption [33]. For clarity, we do not present summary statistics (i.e., 95% CIs and $p$-values) for all individual CpG associations in the main text given the large number of associations present in basic and fully adjusted models. However, these are made available in **S6** and **S8 Tables**, respectively.

Next, we conducted a fully adjusted model that further accounted for 5 common lifestyle risk factors and population structure. The 5 risk factors were alcohol consumption, body mass index, deprivation (Scottish Index of Multiple Deprivation), a methylation-based proxy for tobacco smoking [23], and years of education. The fully adjusted model returned 78 associations across 8 disease states ($p < 2.6 \times 10^{-9}$; **Fig 2B, S8 Table**). Sixty-nine associations from the basic model were also present in the fully adjusted analysis. The 69 associations were spread across 4 disease states: CKD ($n = 1$); ischemic heart disease ($n = 6$); breast cancer ($n = 10$); and type 2 diabetes ($n = 52$). Genomic inflation factors ranged from 0.8 to 1.8 across all fully adjusted models and were 1.1, 1.8, 1.0, and 1.1 for CKD, ischemic heart disease, breast cancer, and type 2 diabetes, respectively (**S7 Table**). The significant findings included associations between self-reported history of breast cancer and hypomethylation within cg06072257 and cg06123699, which are located near *UBIAD1 and TPRG1* on chromosomes 1 and 3, respectively ($p = 6.5 \times 10^{-103}$ and $p = 2.4 \times 10^{-101}$, respectively). The site cg17944885 located between *ZNF788* and *ZNF20* on chromosome 19 associated with prevalent CKD ($p = 1.7 \times 10^{-12}$). Furthermore, CpGs annotated to *ABCG1*, *DHCR24*, and *MYLIP* were common to ischemic heart disease and type 2 diabetes (**Fig 2B**). We also examined where the 69

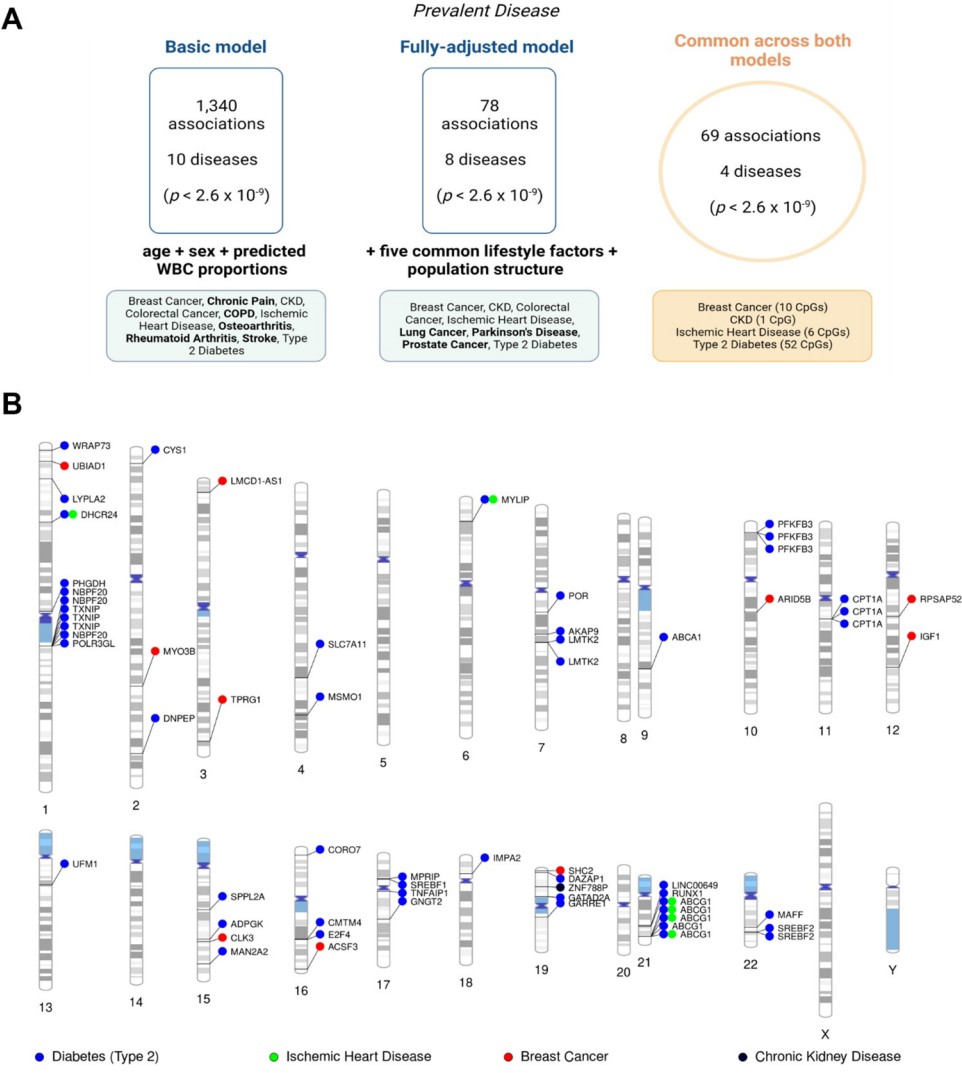

**Fig 2. Epigenome-wide association studies on 14 prevalent disease states in Generation Scotland.** (A) Diseases that had CpG associations in only the basic model or the fully adjusted model are shown in bold. Colorectal cancer was present in both the basic and fully adjusted model, but no CpGs were common to both models for this condition. (B). Ideogram showing 69 sites that were common to both the basic and fully adjusted models. These loci associated with 4 unique disease states. Full information is available in **S8 Table**. Image created using Biorender.com. CKD, chronic kidney disease; COPD, chronic obstructive pulmonary disease; CpG, cytosine-phosphate-guanine dinucleotide; WBC, white blood cells.

associations of interest were located in relation to CpG islands. CpG islands are clusters of methylation sites that typically occur at or near transcription start sites. Only 1 CpG was annotated to a CpG island (cg00994936), 20 were located in shores (0 to 2 kb from islands), 11 were in shelves (2 to 4 kb from islands), and the remaining 37 were annotated to the "open sea" (isolated sites outside of islands) (**S8 Table**).

Genetic colocalisation analyses provided weak evidence for a shared causal variant underlying methylation at cg00857282 (*MYLIP*) and risk of ischemic heart disease (PP = 63%; **S9 Table**). There was also moderate evidence for distinct causal variants underlying 10 of the 69 prevalent associations (PP > 75%).

### 3.3. Epigenome-wide analyses on incident disease

Using health record linkage, we tested whether CpGs measured at baseline associated with the future onset of 19 disease states. We observed 14,237 associations between baseline CpG methylation and the incidence of 11 disease states in the basic model ($p < 1.9 \times 10^{-9}$; **Fig 3A, S10 Table**). Of these, 11,305 (79.4%) and 2,657 (18.7%) were attributed to COPD and type 2

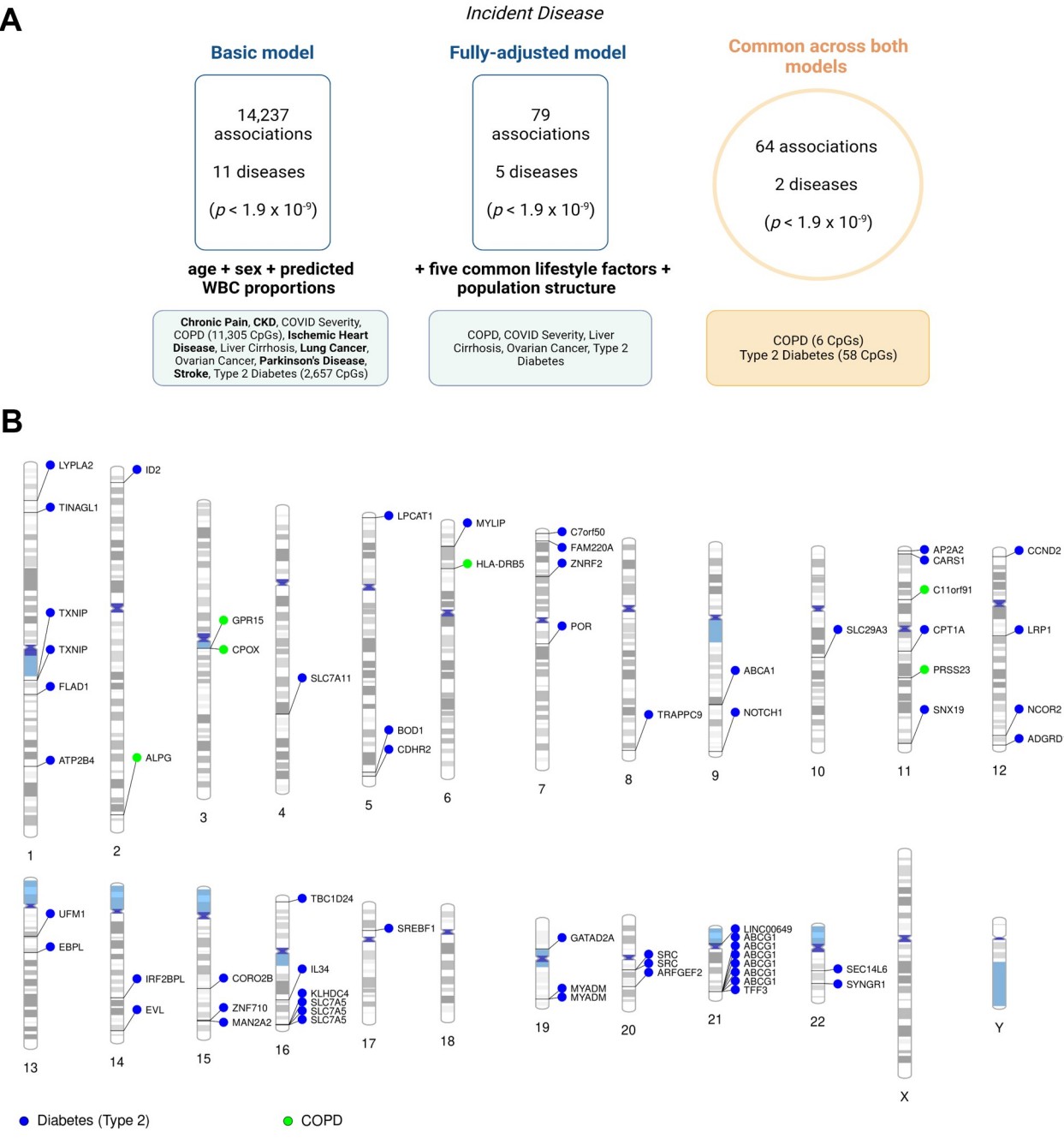

**Fig 3. Epigenome-wide association studies on 19 incident disease states in Generation Scotland.** Diseases that were identified in only the basic model or only the fully adjusted model are shown in bold. COVID severity, liver cirrhosis, and ovarian cancer were present in both a basic and fully adjusted model, but there were no overlapping CpGs for these disease states in both models. (B). Ideogram showing 64 associations that were common to the basic and fully adjusted models. Full information is available in **S12 Table**. Image created using Biorender.com. COPD, chronic obstructive pulmonary disease; CpG, cytosine-phosphate-guanine dinucleotide; WBC, white blood cells.

diabetes, respectively. Well-established smoking-associated probes (e.g., cg14391737 within *PRSS23* and cg05575921 within *AHRR*) associated with the incidence of COPD, lung cancer, ischemic heart disease, stroke, pain, and/or CKD. Genomic inflation factors ranged from 0.8 to 3.8 across all basic incidence models (**S11 Table**).

There were 79 unique associations in the fully adjusted model, which were spread across 5 disease states (**Fig 3B**, **S12 Table**). However, only 64 associations for COPD (*n* = 6) and type 2 diabetes (*n* = 58) were present across both basic and fully adjusted models. One site was annotated to a CpG island (cg14334350), 10 were in shores, 12 were in shelves, and 41 were located in the "open sea." Genomic inflation factors ranged from 0.8 to 1.8 across all fully adjusted incidence models and were 1.1 and 1.8 for COPD and type 2 diabetes, respectively (**S11 Table**). Genes annotated to CpGs that associated with COPD included *ALPG*, *C11orf91*, *CPOX*, *GPR15*, *HLA-DRB5*, and *PRSS23*. Genes annotated to CpGs that were associated with type 2 diabetes included *ABCA1*, *ABCG1*, *CPT1A*, *SREBF1*, *SLC7A11*, *SLC7A5*, and *TXNIP* among others (see **S12 Table** for full details). Only type 2 diabetes had CpGs common to cross-sectional and longitudinal analyses and reflected 17 CpGs annotated to 11 unique genes.

There was only moderate evidence for distinct causal variants underlying 11/64 incident associations (PP > 75%). No associations showed strong evidence of colocalisation (**S13 Table**).

As a further analysis, we examined the contribution of each of the 5 common lifestyle risk factors in attenuating the 1,340 prevalent associations and 14,237 incident associations that were brought forward to the fully adjusted stage. The findings are outlined in full in **S5 Text** and **S14 Table**. In brief, the mean attenuation in effect sizes by each of the covariates ranged from 5.5% (for body mass index) to 63.1% (for smoking). However, there was heterogeneity across disease states given their distinct risk profiles.

### 3.4. Pathway enrichment analysis for methylation sites associated with common disease states

The top 100 CpGs (i.e., with the smallest EWAS *p*-values) for each fully adjusted model were assessed for enrichment in KEGG pathways and GO terms (see Methods). Thirty-three models were considered and reflected 14 prevalent and 19 incident phenotypes (**S15 Table**). Significant pathways were returned only for prevalent type 2 diabetes and ischemic heart disease (FDR-corrected *p*-value <0.05). The overrepresented terms included cholesterol and metabolic processes as well as alcohol metabolic pathways, which may indicate residual confounding despite adjustment for self-reported alcohol consumption.

### 3.5. Associations between CpG methylation and disease states are robust in sensitivity analyses

Mixed-effects models that included a kinship matrix were used to account for relatedness as sensitivity analyses. Effect sizes correlated >0.99 with associations from the standard EWAS, which included related individuals (**S16 and S17 Tables**, **S3 Fig**). Further, Cox proportional hazard models are often used to conduct incidence analyses. This model relies on the proportional hazard assumption, which in effect states that the hazard ratio remains constant over time and implies that the effect of a risk variable is also constant over the length of follow-up. The assumption is supported by a nonsignificant relationship between Schoenfeld residuals and time and refuted by a significant association. Fourteen of the 64 incident associations violated the proportional hazard assumption (*p* < 0.05 between Schoenfeld residuals and time; **S18 Table**). However, we also restricted the analyses to each possible year of follow-up and found that there were minimal differences in hazard ratios between time-points that failed the assumption versus those that did not (**S19 Table**). This suggested the hazards were

proportional over time and all associations were therefore retained. Furthermore, death was considered as a censoring event within our study rather than a competing risk. Effect sizes were correlated >0.99 when incidence models were repeated with death as a competing event, and when individuals who had died were excluded from analyses (**S20 Table**).

The large number of association models employed in EWAS renders it challenging to examine the potential influence of outlying values for each CpG site, particularly where multiple phenotypes are evaluated. In an effort to highlight possible influential outliers, we computed Cook's distance measurements across all 69 prevalent associations (4 prevalent phenotypes) and 64 incident associations (2 incident phenotypes) that were present in basic and fully adjusted models. There were therefore 133 association models for which Cook's distance was computed. Cook's distance is a measure of the effect of deleting an observation on the estimated coefficients, and the associated plots for all 133 models are shown in S2 **Appendix** [34,35]. Two separate criteria were used to identify influential outliers: (i) individuals were deemed as outliers if their distance was greater than 3 times the mean distance across data points (standard rule of thumb) or (ii) a smaller subset of "extreme outliers" were identified based on visual inspection of the plots. There were between 174 to 565 outliers across models using the first criterion and 0 to 4 extreme outliers identified by the second criterion. Effect sizes were correlated 0.7 with those from the original EWAS when outliers from the first criterion were removed and 0.99 when those from the second criterion were omitted (**S21 Table**).

Fully adjusted models were repeated using logistic regression (prevalent disease) or Cox models (incident disease) with age and sex included as fixed-effect covariates. This differs from the main analytical strategy that used linear regression models with adjusted phenotype and methylation variables and allowed us to return effect sizes on an interpretable scale. **Fig 4** shows odds ratios and hazard ratios associated with a per-1 SD increase in adjusted CpG methylation M-values for all 69 and 64 prevalent and incident disease associations (**S22 Table**). We also computed the Harrell's C-statistic for each of the 64 incident associations, which is a measure of goodness of fit within survival analyses. Specifically, we calculated the difference between the C-statistic between a fully adjusted model with and without each CpG of interest. The model without the CpG included age, sex, estimated blood cell proportions, population structure, and 5 common lifestyle factors as outlined previously. The C-statistic from this model was 0.87 and 0.80 for COPD and type 2 diabetes, respectively. All CpGs increased the concordance index. The increment obtained from CpGs ranged from 0.1% to 1.2% (for cg00163198, type 2 diabetes) across all 64 loci (**S22 Table**).

### 3.6. Structured literature review on existing epigenome-wide analyses of common diseases

We performed a structured review of the literature to identify blood-based EWAS on the 19 disease states considered in our study ($n$ = 54 studies; **Fig 5**). Characteristics for each of the 54 studies are outlined (**S23 Table**). The studies were deemed to be of high quality. However, there was a high risk of selection bias among epigenome-wide analyses as well as attrition bias (i.e., in the incidence analyses). Fourteen disease states had at least 1 EWAS reported in the literature. The number of studies ranged from 1 (for long COVID) to 7 (for type 2 diabetes and lung cancer) (**S24** and **S25** **Tables**). Four studies used the Illumina 27k array (7.4%), 36 used the 450k array (66.7%), 12 employed the EWAS array (22.2%), and 2 implemented alternative arrays (Infinium Multi-Ethnic Global-8 and PyroMark Q24, 3.7%). Sixteen studies examined incident disease, while the remaining 38 focused on prevalent disease.

First, we performed look-up analyses to determine whether CpGs identified in our study were previously reported at significance thresholds deemed by each individual study. Only 11/

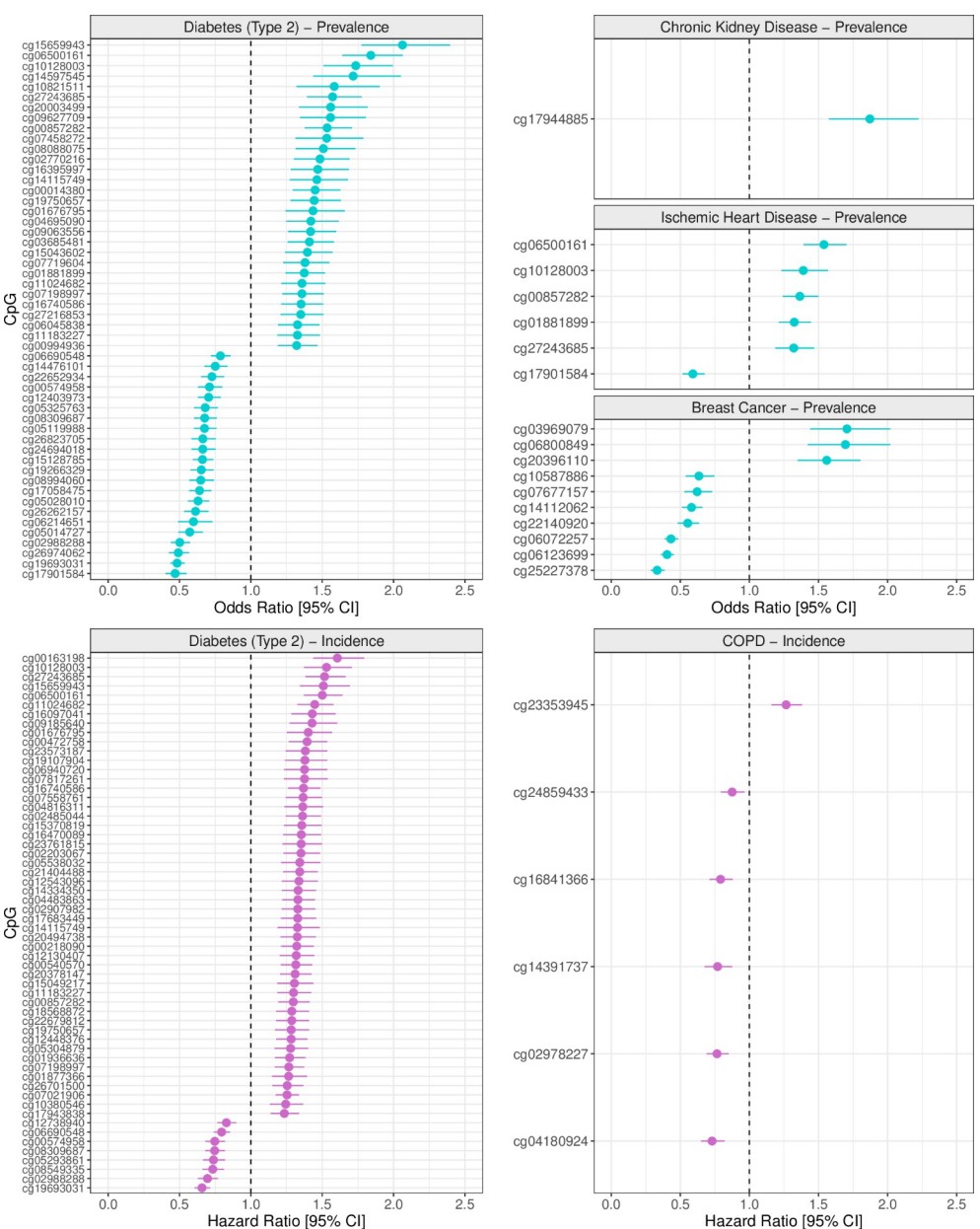

**Fig 4. Blood CpGs associated with prevalent or incident disease states showing effect sizes on interpretable scale.**
Effect sizes were reestimated using logistic regression (prevalent disease, blue points) or Cox proportional hazards models (incident disease, violet points) to return more interpretable effect sizes. Effect sizes represent a per-1 SD increase in age-, sex-, and experimental batch-adjusted CpG methylation M-values (or age- and batch-adjusted for breast cancer). CpGs shown were significant in both basic and fully adjusted models. Odds ratios and hazard ratios are detailed in **S22 Table**. CI, confidence interval; CpG, cytosine-phosphate-guanine dinucleotide; SD, standard deviation.

69 prevalent associations in this study (including 1 for CKD and 10 for type 2 diabetes) and 8/64 incident associations (for type 2 diabetes only) were reported in the literature (at $p < 2 \times 10^{-5}$, which represented the least conservative threshold across studies for these traits; **Fig 5**). The replicated associations for type 2 diabetes implicated genes including *ABCG1*, *CPT1A*, *SREBF1*, and *TXNIP*.

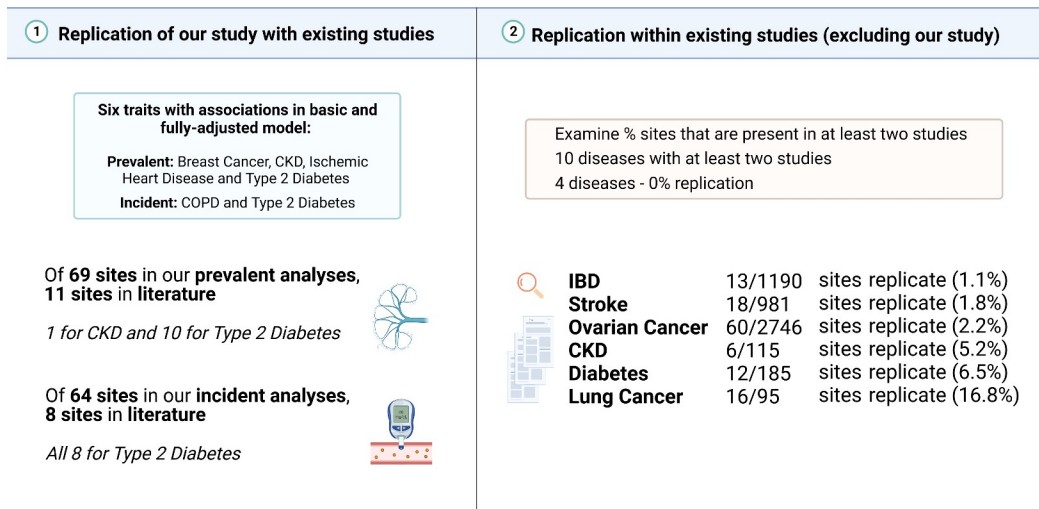

**Fig 5. Look-up and replication analyses within EWAS on common disease states.** A structured literature search was performed to identify existing EWAS on 19 common disease states (either prevalent or incident). (1) We first determined whether associations in our study replicated those of previous studies. We focussed only on associations that were common to basic and fully adjusted models. There were 69 prevalent associations across 4 conditions (breast cancer, CKD, ischemic heart disease, and type 2 diabetes), and 64 incident associations across 2 conditions (COPD and type 2 diabetes). We found that 11/69 prevalent associations and 8/64 incident associations were reported in the literature. (2) We then turned our attention to the existing studies and asked whether studies that examined the same trait (e.g., incident type 2 diabetes) reported the same loci in their studies. We omit our study here as we are only interested in the previous literature. We required that a CpG site was reported in at least 2 studies that examined the same trait. There was a limited amount of replication in the literature as indicated in the right-hand side of the figure. Image created using Biorender.com. CKD, chronic kidney disease; CpG, cytosine-phosphate-guanine dinucleotide; COPD, chronic obstructive pulmonary disease; IBD, inflammatory bowel disease.

Second, we assessed how well previous studies that examined the same trait (e.g., the 7 studies on type 2 diabetes) agreed with one another in terms of locus discovery. The present study was not included in this analysis as here we were interested only in the previous literature. A CpG was considered to be replicated in the literature if 2 or more studies reported it as significant at the threshold defined in their study. As different arrays may not have the same CpG sites, we also considered whether a given gene was replicated in at least 2 studies examining the same condition. There were 10 disease states that were available for testing (i.e., had 2 or more studies with available summary statistic data). The number of unique CpGs that were reported as significant by the authors ranged from 7 (for COPD) to 2,746 (for ovarian cancer). Six of the 10 disease states had evidence of replication across existing studies with respect to the CpGs identified by EWAS. They were IBD (1.1% of CpGs replicated), stroke (1.8%), ovarian cancer (2.2%), CKD (5.2%), type 2 diabetes (6.5%), and lung cancer (16.8%) (**Fig 5**). Similar percentages were observed for genes, with the exception of CKD, which had no common genes across studies as all of the replicated CpGs were intergenic (**S25 Table**).

## Discussion

Using one of the world's largest methylation datasets, we perform a series of EWAS on the prevalence and incidence of a broad range of conditions. We undertook a large-scale, comprehensive review of the literature and highlight the poor agreement that exists across previous epigenome-wide analyses that examine the same condition. By comparing these data with our own findings, we uncover 58 novel associations with the prevalence of 3 self-reported disease states at the study baseline (breast cancer, ischemic heart disease, and type 2 diabetes). We also

identify 56 novel associations between CpGs and the time-to-onset of 2 disease states (COPD and type 2 diabetes). These associations were independent of common lifestyle risk factors. However, we also observe a vast number of additional associations whereby CpGs index or track associations between lifestyle factors and common disease states, further highlighting the appropriateness of DNAm as a biomarker of lifestyle behaviours.

The novel associations observed in this study could strengthen evidence for candidate molecular pathways underlying peripheral disease states, e.g., self-reported history of breast cancer associated with differential methylation at cg06072257 (*UBIAD1*) and cg06123699 (*TPRG1*). UBIAD1 (UbiA Prenyltransferase Domain Containing 1) is a biosynthetic enzyme that converts vitamin K1 (phylloquinone) to menaquinone, which is the most abundant form of vitamin K2 in human tissue [36]. Low expression of UBIAD1 in human breast tumours correlates with reduced survival [37] and also associates with risk for bladder cancer [38]. *TPRG1* encodes for Tumour protein P63 Regulated 1 and its expression is associated with estrogen receptor-positive and triple-negative breast cancers [39,40]. Furthermore, in relation to COPD, cg23353945 (*C11orf91*) correlated with incidence of the disease and has been associated in *trans* with CCL21 protein levels [41]. Serum CCL21 levels are elevated in COPD patients and may contribute to the development of lung cancer [42,43]. This may suggest that a C11orf91-CCL21 axis contributes to risk of pulmonary disease independently from lifestyle risk factors. However, these findings warrant further investigation in mechanistic in vitro and in vivo studies.

The most consistent associations across models and look-up analyses were for type 2 diabetes. This is likely attributed to the strong correlation between metabolic processes (e.g., glucose and lipid metabolism) and DNAm in blood [44]. The condition with the highest degree of replication within the existing literature alone was lung cancer. This may reflect the strong influence of smoking on DNAm. From these analyses, it is apparent that EWAS possess a general low level of replicability, in particular when compared to genome-wide association studies (or GWAS), which show replication rates of 50% to 90% [45,46]. However, unlike DNAm, genetic factors remain fixed across the life-course and large sample sizes in GWAS have ensured adequate power. Epigenetic analyses are also highly susceptible to adjustments for environmental exposures as indicated above. Caution should be paid to covariate strategies particularly where the primary objective is to identify causal molecular mechanisms that connect genetic risk to disease endpoints, which should mandate high replicability. Furthermore, in our study, EWAS were conducted using linear regression models, which examined each CpG site in isolation. The risk of overfitting was low due to the large number of observations compared to the number of model parameters. However, the vast number of associations observed in our analyses may be attributable to the large sample size and possibly to the correlation structure among CpG sites within the same genomic region or distal sites influenced by similar lifestyle factors. As sample sizes grow, it may be necessary to employ additional methods that permit the joint and conditional estimation of probe effects while accounting for correlation structure and unknown confounders [20,47].

The generally poor replication across existing EWAS reflects a number of possible factors. These include the use of (i) different statistical models and significance thresholds; (ii) arrays with different CpG content (e.g., 450k versus EPIC arrays); (iii) different study designs (e.g., community-based designs with no enrichment for a particular disease versus targeted case/control designs); (iv) heterogeneities in genetic backgrounds; (v) variation in phenotype definitions for health record linkage analyses; and (vi) the use of disparate covariate strategies. Some studies also did not make full summary statistics available. Nevertheless, our review is critical and timely given that the scale of EWAS continues to rise in tandem with enhancements in array technologies, population biobank sizes, and health record phenotyping algorithms.

We highlight a number of further considerations in addition to those arising from the structured literature review. First, there was limited overlap between methylation sites identified in the prevalence and incidence analyses. Prevalence analyses relied on self-report data, which may have been prone to recall bias, whereas incidence analyses considered diagnosed disease. A subset of controls within the prevalence analyses will also have been reassigned to cases in the incidence analyses, which could attenuate common signal between these analyses. Second, the majority of disease states showed weak associations with differential methylation at CpG sites despite the large sample size employed. This is further highlighted by the lack of consistency in coefficient estimates across models. It is important to note that while the overall sample size was large, the number of cases in many conditions was modest, which may have limited power. The analyses also emphasise that epigenome-wide analyses are highly sensitive to adjustments for environmental exposures. Third, colocalisation analyses did not provide evidence that altered methylation and disease risk mechanisms shared common genetic variants. The CpG associations may instead reflect distinct genetic aetiologies, unknown confounding factors, and some of the associations could capture subclinical disease in the participants. Fourth, we did not consider multimorbidity in this study. There are a number of possible trajectories that a particular participant may have shown, as well as a number of recorded events for a given condition (e.g., stroke). Indeed, we focused on time-to-first-event in this study alone. Future research will focus on applying sophisticated statistical methods to model all possible multimorbidity trajectories from linked healthcare data and disentangle their relationships with peripheral methylation.

Our study has a number of limitations. First, winsorization of methylation values was not applied in our study. Winsorizing limits extreme values in the data, e.g., in M-values for a given CpG site, and can reduce the effect of possibly spurious outliers [34]. However, sensitivity analyses using Cook's distance metrics suggested that regression coefficients were largely stable when influential data points were removed, particularly where extreme outliers were excluded. Second, we did not adjust for medication data, which may confound associations between peripheral methylation and disease. Third, we did not consider disease subtypes as this may have reduced power to detect associations. Fourth, we utilised family history of Alzheimer's disease as proxy for prevalent disease due to the young mean age of the sample at baseline. This complicates its generalisability with incident analyses on Alzheimer's disease, which relied on diagnosed disease. Our phenotype definitions may also have neglected potential cases for other disorders such as CKD, including individuals with proteinuria and normal eGFR or with tubular disorders. Indeed, there is stark heterogeneity in clinical presentations among all conditions considered in our study given their multifactorial aetiologies. Future research may benefit from focussing on precise common endpoints in the disease process, such as fibrosis for CKD and liver cirrhosis. Fifth, our findings in blood might not reflect important changes in distal, disease-relevant tissues. Sixth, our analyses consisted of individuals with European ancestry and might not be generalisable to individuals of other ancestries. Seventh, the look-up analyses in our structured literature review relied on genome-wide significant $p$-value thresholds set by individual studies. This metric is not fully informative given that significant associations will be tightly coupled to characteristics such as the sample size of the study.

Moving forward, we recommend that studies examining the same condition could engage in consortium efforts, which may provide an opportunity to reach consensus on covariate strategies and normalisation methods. Furthermore, it is essential that all studies report clearly the output of nested models, such as models with and without adjustments for lifestyle risk factors, and provide full publicly available summary statistics where possible.

Our epigenome-wide analyses uncovered over 100 novel associations between blood CpGs and common disease states that act independently of major confounding risk factors. Our summary data and synthesis of the literature provide a timely foundation that will expedite discoveries into the role of blood DNAm in common disease states.

## Supporting information

**S1 STROBE Checklist. STROBE statement—Checklist of items that should be included in reports of observational studies.**
(DOCX)

**S1 Appendix. Disease code lists.**
(XLSX)

**S2 Appendix. Cook's distance plots for 133 associations in basic and fully adjusted models.**
Outliers are highlighted in green (COPD) and blue (type 2 diabetes).
(PDF)

**S1 Text. Supplementary methods for methylation quality control.**
(DOCX)

**S2 Text. Supplementary methods for preparation of phenotypes.**
(DOCX)

**S3 Text. Supplementary methods for sensitivity EWAS.**
(DOCX)

**S4 Text. Supplementary methods for methylation QTL analyses.**
(DOCX)

**S5 Text. Supplementary note on covariate-specific attenuation of effect sizes in basic model.**
(DOCX)

**S1 Fig. Associations between covariates and prevalent disease states in univariable and multivariable logistic regression models.**
(DOCX)

**S2 Fig. Associations between covariates and incident disease states in univariable and multivariable Cox proportional hazards models.**
(DOCX)

**S3 Fig. Correlation between effect sizes from linear regression EWAS and sensitivity linear mixed effects analyses that further accounted for relatedness.**
(DOCX)

**S1 Table. Summary data for demographic variables and covariates.**
(XLSX)

**S2 Table. Counts for prevalent disease states.**
(XLSX)

**S3 Table. Counts for incident disease states.**
(XLSX)

**S4 Table. Associations between covariates and prevalent disease states.**
(XLSX)

**S5 Table. Associations between covariates and incident disease states.**
(XLSX)

**S6 Table. Significant associations from basic model—Epigenome-wide association studies on prevalent disease states.**
(XLSX)

**S7 Table. Genomic inflation factors for epigenome-wide association studies on prevalent disease states.**
(XLSX)

**S8 Table. Significant associations from fully adjusted model—Epigenome-wide association studies on prevalent disease states.**
(XLSX)

**S9 Table. Genetic colocalisation analyses for prevalent disease associations.**
(XLSX)

**S10 Table. Significant associations from basic model—Epigenome-wide association studies on incident disease states.**
(XLSX)

**S11 Table. Genomic inflation factors for epigenome-wide association studies on incident disease states.**
(XLSX)

**S12 Table. Significant associations from fully adjusted model—Epigenome-wide association studies on incident disease states.**
(XLSX)

**S13 Table. Genetic colocalisation analyses for incident disease associations.**
(XLSX)

**S14 Table. Sensitivity analysis to test for the effects of each of the 5 lifestyle risk factors included in this study on attenuating associations from the basic model.**
(XLSX)

**S15 Table. Pathway enrichment analyses.**
(XLSX)

**S16 Table. Sensitivity analysis to test for effect of relatedness on associations with prevalent disease states.**
(XLSX)

**S17 Table. Sensitivity analysis to test for effect of relatedness on associations with incident disease states.**
(XLSX)

**S18 Table. Sensitivity analysis to test for proportional hazard assumption.**
(XLSX)

**S19 Table. Sensitivity analysis to estimate hazard ratios during each year of follow-up for associations that violated proportional hazard assumption in S18 Table.**
(XLSX)

**S20 Table. Sensitivity analysis to assess the impact of all-cause mortality as a competing risk in incidence models.**
(XLSX)

**S21 Table. Sensitivity analysis to identify influential observations based on Cook's distance statistics.**
(XLSX)

**S22 Table. Odds ratios and hazard ratios for prevalent and incident disease associations, including Harrell's C-statistic for the latter.**
(XLSX)

**S23 Table. Characteristics of 54 studies identified in structured literature review.**
(XLSX)

**S24 Table. Look-up analyses to assess whether associations identified in the present study are newly described.**
(XLSX)

**S25 Table. Replication within existing epigenome-wide association studies that examined the same condition in the literature.**
(XLSX)

## Author Contributions

**Conceptualization:** Robert F. Hillary, Riccardo E. Marioni.

**Data curation:** Daniel L. McCartney, Lee Murphy, Nicola Wrobel, Archie Campbell, Rosie M. Walker, Caroline Hayward, Kathryn L. Evans, Andrew M. McIntosh.

**Formal analysis:** Robert F. Hillary, Daniel L. McCartney, Elena Bernabeu, Danni A. Gadd.

**Funding acquisition:** Robert F. Hillary, Caroline Hayward, Kathryn L. Evans, Andrew M. McIntosh, Riccardo E. Marioni.

**Investigation:** Robert F. Hillary, Riccardo E. Marioni.

**Methodology:** Robert F. Hillary, Daniel L. McCartney, Elena Bernabeu, Danni A. Gadd, Riccardo E. Marioni.

**Project administration:** Riccardo E. Marioni.

**Software:** Robert F. Hillary, Daniel L. McCartney, Hannah M. Smith, Aleksandra D. Chybowska, Yipeng Cheng.

**Supervision:** Riccardo E. Marioni.

**Visualization:** Robert F. Hillary.

**Writing – original draft:** Robert F. Hillary, Riccardo E. Marioni.

**Writing – review & editing:** Robert F. Hillary, Daniel L. McCartney, Hannah M. Smith, Elena Bernabeu, Danni A. Gadd, Aleksandra D. Chybowska, Yipeng Cheng, Lee Murphy, Nicola Wrobel, Archie Campbell, Rosie M. Walker, Caroline Hayward, Kathryn L. Evans, Andrew M. McIntosh, Riccardo E. Marioni.

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
