## [Editor Report · Decision Letter 0]

10 Jan 2023

Dear Dr Hillary, 

Thank you for submitting your manuscript entitled "Blood-based epigenome-wide analyses on the prevalence and incidence of nineteen common disease states" for consideration by PLOS Medicine.

Your manuscript has now been evaluated by the PLOS Medicine editorial staff as well as by an academic editor with relevant expertise and I am writing to let you know that we would like to send your submission out for external peer review.

Please re-submit your manuscript within two working days, i.e. by Jan 12 2023 11:59PM.

Kind regards,

Philippa Dodd, MBBS MRCP PhD

PLOS Medicine

---

## [Decision Letter · Decision Letter 1]

4 Mar 2023

Dear Dr. Hillary,

Thank you very much for submitting your manuscript "Blood-based epigenome-wide analyses on the prevalence and incidence of nineteen common disease states" (PMEDICINE-D-22-04026R1) for consideration at PLOS Medicine. 

Your paper was evaluated by a senior editor and discussed among all the editors here. It was also sent to independent reviewers, including a statistical reviewer. The reviews are appended at the bottom of this email and any accompanying reviewer attachments can be seen via the link below:

[LINK]

In light of these reviews, I am afraid that we will not be able to accept the manuscript for publication in the journal in its current form, but we would like to consider a revised version that addresses the reviewers' and editors' comments. Obviously we cannot make any decision about publication until we have seen the revised manuscript and your response, and we plan to seek re-review by one or more of the reviewers. 

We expect to receive your revised manuscript by Mar 27 2023 11:59PM. Please email us (plosmedicine@plos.org) if you have any questions or concerns.

We look forward to receiving your revised manuscript. 

Sincerely,

Philippa Dodd, MBBS MRCP PhD

PLOS Medicine

plosmedicine.org

GENERAL

Please respond to all editor and reviewer comments detailed below in full.

Please number the lines starting at 1 and in continuous sequence throughout, thereafter.

Your study combines a cohort study and a systematic review/meta-analysis (SRMA). Each of these study designs has its own reporting guidance. Please review both the STROBE guideline (cohort studies) and the PRISMA guideline (SRMAs) and report the relevant parts of your study according to the guidance.

Please include the completed STROBE checklist as Supporting Information. Please add the following statement, or similar, to the relevant section of the Methods: "This study is reported as per the Strengthening the Reporting of Observational Studies in Epidemiology (STROBE) guideline (S1 Checklist)." Please use section and paragraph numbers, rather than page or line numbers as these often change at the time of publication.

The PRISMA guideline can be found here: https://www.equator-network.org/reporting-guidelines/prisma/

Please provide the completed PRISMA checklist as supporting information. 

When completing the checklist, please use section and paragraph numbers, rather than page or line numbers as these often change at the time of publication. Please add the following statement, or similar, to the relevant section of the Methods: "This study is reported as per the Preferred Reporting Items for Systematic Reviews and Meta-Analyses (PRISMA) guideline (S1 Checklist)."

PLOS Medicine appeals to a wide general medical audience, to whom we think your study would appeal. To make your manuscript widely accessible we suggest ensuring clear use of language and particular attention to the explanation and definition of terms that are likely to be less familiar to the more general reader, as opposed to the epigeneticist. We also suggest caution with your use of causal language and claims of primacy.

TITLE

Please revise your title according to PLOS Medicine's style. Your title must be nondeclarative and not a question. It should begin with main concept if possible. "Effect of" should be used only if causality can be inferred, i.e., for an RCT. Please place the study design ("A randomized controlled trial," "A retrospective study," "A modelling study," etc.) in the subtitle (ie, after a colon).

ABSTRACT

Please structure your abstract using the PLOS Medicine headings (Background, Methods and Findings, Conclusions).

Please combine the Methods and Findings sections into one section, “Methods and findings”.

Abstract Background: 

Please provide the context of why the study is important. 

The final sentence should clearly state the study question.

“…epigenome-wide analyses…” suggest defining as “(EWAS)” here. It may also be helpful to further elaborate on the importance and potential uses of epigenetic data

Abstract Methods and Findings:

Please include further details of the study population and setting, number of participants, years during which the study took place, length of follow up, and main outcome measures.

What health record database did you link the epigenetics data to?

Please define “CpGs” at first use in the abstract.

“We identify 69 associations between CpGs and the prevalence of four disease states” Before you present the results some further brief details of what you did would be helpful - What diseases did you investigate? what associations did you investigate? How did you decide which diseases to investigate? 

Please provide additional details of your literature review – dates of search, data sources, number of studies included, types of study designs included, eligibility criteria, and synthesis/appraisal methods

Please quantify the main results with 95% CIs and p values. When reporting p values please report as p<0.001 or where higher as p=0.002, for example

Please ensure that all numbers presented in the abstract are present and identical to numbers presented in the main manuscript text.

Please include any important dependent variables that are adjusted for in the analyses.

In the last sentence of the Abstract Methods and Findings section, please describe the main limitation(s) of the study's methodology.

Abstract Conclusions:

Please address the study implications without overreaching what can be concluded from the data; the phrase "In this study, we observed ..." may be useful.

Please interpret the study based on the results presented in the abstract, emphasizing what is new without overstating your conclusions.

Please avoid vague statements such as "these results have major implications for policy/clinical care". Mention only specific implications substantiated by the results.

Please avoid assertions of primacy ("We report for the first time....")

AUTHOR SUMMARY

The summary should include 2-3 single sentence bullet points under each individual question. We encourage you to review some published articles on our website for examples here https://journals.plos.org/plosmedicine/

INTRODUCTION

Please further explain the need for and potential importance of your study. If there has been a systematic review (other than that you have conducted for this study) of the evidence related to your study, please refer to and reference that review and indicate whether it supports the need for your study.

“…To date, no study…” claims of supremacy can be risky suggest “to our knowledge” or similar

METHODS and RESULTS

Please ensure that your methods section includes details of the study cohort without the need to refer to another article (basic demographics, age, sex, enrollment criteria etc). Please include these details in section labelled 2.1.

Did your study have a prospective protocol or analysis plan? Please state this (either way) early in the Methods section.

For all observational studies, we request that in the manuscript text, authors please indicate the following: 

(1) the specific hypotheses you intended to test, 

(2) the analytical methods by which you planned to test them, 

(3) the analyses you actually performed, and 

(4) when reported analyses differ from those that were planned, transparent explanations for differences that affect the reliability of the study's results. If a reported analysis was performed based on an interesting but unanticipated pattern in the data, please be clear that the analysis was data-driven.

Please ensure that the main results are quantified with 95% CIs and p values. Please report p as p<0.001 and where higher as p=0.002, for example. For the purpose of transparent data reporting, if not please clearly state why not.

LITERATURE REVIEW

Please move the details of your literature review to the main manuscript. Such that the following is included dates of search, data sources, types of study designs included, eligibility criteria, and synthesis/appraisal methods. 

We require that SRs are updated to within roughly 6 months of the expected publication date. Please update your search to the present time. We also ask for an evaluation of study quality and risk of bias and for an evaluation for evidence of publication bias. Please include.

TABLES

Please include a table containing the baseline characteristics of your study population

FIGURES

Please ensure that CpG is defined within the figure captions where relevant, figure 1, for example.

To help facilitate transparent data reporting, PLOS Medicine requests that where adjusted analyses are presented unadjusted analyses are presented for comparison. Please include unadjusted analyses. If not including unadjusted analyses, then please clearly state the reasons why not.

The + and and * as well as the text in part B of the figures are very difficult to read even with the figure enlarged, please revise to improve accessibility to the reader

Please consider avoiding the use of red and/or green to improve accessibility of your figures to those with color blindness

Please quantify the main results with 95% CIs and p values. Please report p values as p<0.001 or where higher as p=0.002, for example. For the purpose of transparent data reporting, if not please clearly state why not.

FIGURE 5: please revise the statement “our study is the first with”

DISCUSSION

Please present and organize the Discussion as follows: a short, clear summary of the article's findings; what the study adds to existing research and where and why the results may differ from previous research; strengths and limitations of the study; implications and next steps for research, clinical practice, and/or public policy; one-paragraph conclusion. Please ensure that the discussion reads as a single piece of continuous prose without any sub-headings.

Please remove the sub-heading “Conclusions”

Please move the ethics statement to the methods section of the main manuscript.

Please remove data availability, funding and competing interest statements from the end of the manuscript and include only in the manuscript submission form when you resubmit.

REFERENCES

Please ensure that in-text reference callouts are placed within square parentheses preceding punctuation, as follows, “For example [1,3,6].” Please note the presence of a space preceding the opening parenthesis and the absence of spaces between citations.

In your bibliography (including in the supporting files) please ensure that up to but no more than 6 author names are listed followed by et al., where more than 6 authors contribute to an individual study.

Journal name abbreviations should be those found in the National Center for Biotechnology Information (NCBI) databases. 

Please see our website for further reference guidelines https://journals.plos.org/plosmedicine/s/submission-guidelines#loc-references

SUPPORTING FILES

Please include the PRISMA and STROBE checklists, as detailed above

Please apply the same suggestions above for tables and figures to those in the supporting files as relevant.

Comments from the reviewers:

Reviewer #1: Hilary et al report the results of a phenome - wide EWAS analysis of in Generation Scotland (N≤18,413). Using Illumina EPCI methylation arrays association of individual CpGs were assessed with prevalence of 14 disease states at base line (mean age 47.5) and incidence of 19 disease states over follow up (~9-14 years).

Association analysis was carried out using linear regression models were used for EWAS via the OSCA (OmicS-data-based Complex trait Analysis) software. Basic model (houseman cell type proportions) and adjusted (cell types. Genetic principal components and lifestyle factors) models were run. Cox proportional-hazards models were used for survival analysis for incidence phenotypes

Adjustment was for number of CpGs + number of phenotypes. (14 logistic regression, 19 for incidence analysis). M values were used.

The methods for DNA methylation data processing and adjustment for age / sex are appropriate. The level of significance adjusts for the number of phenotypes examined. Sensitivity analyses were carried out to assess the effect of relatedness between subjects in the cohort.

In total 69 associations between CpGs and the prevalence of four disease states at baseline ( 58 are novel). 64 CpGs were associated with the incidence of two disease states (COPD and type 2 diabetes (56 are novel).

Strengths of the study include the sample size adjustment fro lifestyle factors. Weaknesses include the lack of genomic inflation analysis to assess if there is residual confounding and the self report and EPR nature of phenotyping. 

Main points

1. Increasingly winzorisation of data is being adopted in EWAS to reduce the influence of outlier probes. Was this considered?

2. What was the genomic inflation of the models?

3. When determining whether associations were novel was the EWAS atlas used (https://ngdc.cncb.ac.cn/ewas/atlas)? What was the criteria for novel? Unique CpG or unique CpG at unique genomic location?

4. Did the authors consider incorporating enrichment analyses to look at enriched & depleted genomic locations, enriched Go & KEGG terms, related traits in EWAS Atlas for say top 100 CpGs for each analysis? This could give insight into whether these CpGs have previously been corelated with specific environmental exposures for example.

5. It would have been nice to see what the clinical significance of significantly associated CpGs was - how predictive of new onset disease for example?

6. It could be acknowledged in the discussion that for the incidence disease that CpGs might not be on the causal pathway and could reflect sub clinical disease.

Minor points

1. Mean age and average length of follow up could be included in figure 1 legend for clarity to reader.

The lifestyle factors adjusted for should be explicitly stated in manuscript, not just providing a reference

Reviewer #2: The present study characterizes epigenome-wide associations between DNA methylation (DNAm) patterns derived from peripheral blood and both the prevalence and incidence of a range of different diseases, based on data from the Generation Scotland study. The authors provide results from both a core model and a fully-adjusted model controlling for several lifestyle factors, and perform a literature review of published EWASs to investigate the extent to which findings replicate. The authors also conduct co-localization analyses to test whether the top DNAm loci and associated traits are linked to common vs distinct genetic variants. The manuscript is very well-written, it addresses an important topic and has multiple strengths, including the investigation of both disease prevalence (cross-sectional analyses) and incidence (longitudinal analyses) across a broad range of diseases and the use of one of the largest epigenetic datasets in the world. Overall, I believe this is an excellent study and poised to make a significant contribution to the field. Specific comments that could be addressed to strengthen the manuscript are provided below. 

* Abstract

o The authors could also mention in the background that previous EWAS studies typically focus on single diseases (as opposed to the wide range of outcomes examined here)

o Could you provide some key information, such as the age range of the sample, the fact that this is a family-structured study and the prevalence/incidence range across diseases examined. 

o It would be worth stating a bit more explicitly that, based on this large sample, most diseases show rather weak associations (e.g., as indicated by the lack of [consistent] EWAS-significant associations across models)

* Introduction: in the last sentence, the authors state that colocalization analyses are performed to determine 'whether CpG methylation…causally associate with disease risk'. Can the authors clarify whether this is indeed what the colocalization analyses indicate? If both a CpG site and a trait are influenced by the same genetic variant, does it mean that the CpG site is causally involved in the disease, or could it be that the genetic variant has a pleiotropic effect on the CpG and disease, without the CpG being necessarily on the pathway? 

* Figure 1 is very clear, but it would be helpful to list which diseases were included in the prevalence vs incidence analyses (or both). 

* Methods

o The prevalence analyses are based on self-report disease status whereas the incidence analyses are based on linkage with health records. Can the authors comment on how comparable/concordant these measures are, and to what extent this may also contribute to differences in findings between incidence vs prevalence analyses (with only Type 2 diabetes showing some overlap between these EWASs)

o In the methods, the authors describe the family structure of the Generation Scotland cohort and how methylation data was available in three different sample sets with different familial/relatedness characteristics. It was unclear to me from the methods though how this complex kinship structure was taken into account in the analyses. Later in the results section the authors state that sensitivity analyses were performed using mixed-effects models to account for family relatedness but I would suggest making this clearer earlier in the manuscript. With regards to the three analytical sets, I was also unsure whether these were analysed separately and results pooled via meta-analysis (given differences between the sets in terms of selection criteria and also quality control procedures for example between set 1 vs set 2 and 3), or whether this was treated as a single analytical sample?

o Section 2.3. The format for the numbering (x) of the diseases here resembles that used for the references which is a bit confusing. 

o Section 2.6. The number of eligible articles after inclusion criteria was 56, out of 2000 articles identified in the search. This is a very large reduction - can the authors mention some of the main reasons for articles dropping out? Perhaps I missed it in the supplementary but it would be helpful to add a table listing the articles included and key characteristics. 

* Results

o Section 3.1. I appreciate that the authors provide information on the prevalence and incidence of the diseases in the supplementary materials due to space restrictions, but could the authors (i) indicate in the text the range of prevalence/incidence of the diseases in this sample, and (ii) add percentages in the supplementary table in addition to number of cases/controls? 

o Section 3.5. I understand it is not straightforward to establish whether findings between EWAS studies replicate, particularly when full summary statistics may not be provided. My two main concerns with the strategy taken though are that (i) focusing on genes themselves excludes the possibility of testing whether e.g. intergenic CpG sites, which may still be functionally relevant, replicate, as they are not annotated to genes; and (ii) that using a genome-wide significant p-value threshold may be only partially informative, as it depends on sample characteristics such as power. I wonder whether the authors could also utilize other estimates to assess replication based on summary statistics, such concordance in the direction of associations and correlations between effect sizes. 

* Discussion: The discussion is clear and concise, but could be expanded to cover a few more important themes emerging from the paper, including:

o The findings indicate that most diseases seem to show rather weak associations with DNAm, even when using such a large methylation dataset

o Type 2 diabetes emerges as one of the diseases with the strongest/most consistent associations (with some convergence across core/extended models and across prevalence/incidence analyses) 

o Several sites are only significant in fully adjusted models - how do the authors interpret this? 

o Commenting on the findings, meaning and implications of the colocalization analyses (i.e. these inform about shared/distinct genetic effects on DNAm and disease, but do they also inform about the [lack of ]causal effects of DNAm on the disease?).

Reviewer #3: This study seeks to identify epigenetic signals of disease prevalence or incidence in the Generation Scotland study. The paper is concisely written and reports some novel EWAS which is a significant contribution to the existing literature. A particularly interesting feature of the paper is the effort to examine concordance of previous EWAS results but this needs to be strengthened to help the reader gain an informed insight into what these results show.

The concordance between the EWAS in GS and previous literature is not currently very detailed. For example, it is not clear what the authors accept as replication between studies - for disease states where there is a number of EWAS (eg T2D) are they accepting CpG sites that are replicated across all studies or do they accept any CpG that is reported in 2 or more studies? Do the authors consider direction of effect or potential heterogeneity across genetic ancestries (although most studies are predominantly European ancestry)? Do they consider the statistical models used in different EWAS? What is the replication in studies using GS data eg Bermingham et al PMID 30935889?

The information being reported here is of consequence as it tells us if EWAS is an avenue of research worth following at all - if concordance between studies is very low and it is not explainable it suggests there is a problem in running these kinds of studies. It would also help if this was put into context - how much replication do we see between well powered GWAS studies? The other comment here is that the authors do not seem to present an appraisal of overlap amongst previous studies on the same disease phenotypes even though they say this will be presented in the introduction.

I have some minor areas needing clarification or improvement:

In 2.4 the adjustment for age, sex and batch could be more clearly described. How are age/sex/batch adjusted M values generated? What are the batches? The batch correction is likely to be incomplete if only one batch factor was included and this needs to be acknowledged. Other commonly used approaches such as SVA may not be feasible in this analysis model but the authors need to justify their approach as it may impact how they interpret replication with other studies.

In the fully adjusted model, can the authors justify their model choice? For example, is adjustment for BMI appropriate in T2D model?

For the mQTL analyses, did the authors check if the mQTL associations were similar between GS and GoDMC for the CpGs that were present on the 450k array? This would give an estimate of how well powered GS is to detect mQTLs within GS and whether there is substantial heterogeneity between GS and GoDMC mQTLs.

Figure 5 is very difficult to interpret - For example in group 4 (EPIC array, n=2) is this prevalent and/or incident disease? For replication with existing studies is "genes replicate" any CpGs at an annotated locus or something else? For "Replication of our study with existing studies" which of the numerator /denominator are from the current study? Why is only CKD and T2D in this box?

The study lists a number of limitations but these could be discussed in more detail if word count constraints allowed. For example, discuss case ascertainment issues of using parental history of AD as a proxy for variable for AD. The discussion of replication between studies could also be discussed in more detail if space allowed.

Reviewer #4: The manuscript entitled "Blood-based epigenome-wide analyses on the prevalence and incidence of nineteen common disease states" by Hillary et al conducts a large and well powered epigenome-wide association analysis of several disease traits using both a cross-sectional and lontiudinal approach. They identify 69 significant associations in four out of the nineteen disease states. They additionaly identify 64 CpGs that colocalise with two diseases, COPD and Type 2 diabetes of which 56 they consider novel and independent of the five lifestyle factors they include. They also not the poor replication in the majority of previous EWAS and can only replicate these finding in 4/19 diseases investigated. Overall, I enjoyed reviewing the article and found if of high interest, however, I have some constructive comments that I hope the authors will find improve the manuscript.

1) The first of these is no replication of their findings in an indpendent cohort. I suppose given that this is one of the many criticisims of EWAS, I am curious as to why these authors chose not find a suitable replication cohort. I understand that they tried to use the existing literature and EWAS catalog but not all of the nineteen disease states they investigated are well studied in the literature. So here are their findings being somehow biased by the amount of DNA methylation work that has been done in cancer versus other more hetergenous diseases with smaller numbers of research done. 

2) In their intiitial regression they identify 1,340 associations versus 78 in the fully adjusted model. Did they look to see which of these addded covariates in their models may be accounting for this drop in signal? Does this represent some sort of environmental influence on the CpG associated in the basic model.

3) I was a little confused by colocalisation mQTL analyses. It was unclear if they were differentiating between these and did they look at GWAS based SNPs or only use GoDMC and their own GWAS data for these? This question comes from their results, which shows most of those CpGs that are colocalising with disease coming from Generation Scotland versus GoDMC. Is this because of the difference in array used.

4) Also, were the 20 PCs used in the model from GWAS data or from the EPIC data? I'm assuming GWAS but this wasn't clearly laid out in the manuscript. 

5) It would be extremely useful to know where in relation to gene these CpGs are located. Are they in the TSS or are they in islands - this may give some insight into the biology.

6) I really liked the paragraph in the discussion in regards to standardizing practice for EWAS, especially in meta-analysis. I guess my suggestion here would be for some more specific recommendations, so normalisation methods, handling of batch effects, population strucuture, etc. 

Minor comments

In the abstract the first paragraph of the results section says 14 common disease states but everywhere else it is 19. I figure this is a typo.

In the methods you mention five estiamted WBC cell types but in all the models this is six. I know there is some question around colinearity and these cell types but not sure if this was just an error. 

In the abstract you mention CpGs - given the more general medical audience for this journal feel it would be better keep the language more general.

Will these summary data be placed into the EWAS catalogue?

Reviewer #5: In this work, the authors describe an epigenome-wide association study (EWAS) on a cohort of individuals from Generation Scotland. In consideration were 14 baseline self-reported common disease states and also the incidence of 19 disease states inferred by utilising health record data. The study design is split into "Prevalence analysis", a logistic regression approach for the 14 self-reported baseline disease states and "Incidence analysis", a censored Cox survival analysis for the 19 health record inferred disease states.

The study considered 18,413 individuals across a set of 752,722 filtered CpG sites on the Illumina MethylationEPIC array. In total, 69 associations were identified across 4 disease states, of which 58 are novel. Also, a total of 64 CpG sites associate with both COPD and diabetes. The authors also undertook a literature analysis and compared the results of this very large Generation Scotland study with the literature. They find poor replication.

Comments

=======

This very large study is a welcome addition to the EWAS literature. It examines not just self-reported disease state at baseline, but increases the value of the epigenetic data in the study by making use of health records to infer the diagnosis of disease states over time. The authors also take the time to compare the results in the context of the current literature and comment on the degree of replication. I agree with the authors that this review is a critical and timely analysis and commentary.

The inclusion of the basic and fully-adjusted model in the paper is useful, and the large reduction in the number of significant CpG sites after covariate adjustment illustrative of the importance of such adjustment. Figure S1 and S2 and also very illustrative for demonstrating the difference between correlation and causation.

To address:

* The fully-adjusted model needs more discussion around the covariates.

1) For body mass index, there is typically a long tail and these severely obese individuals can place substantial leverage on the regression. It is unclear from the manuscript or Supplementary methods whether BMI or BMI z-scores were used in the model.

2) For alcohol consumption or deprivation index, were these also regularised or normalised in some way?

* For the methylation-predicted WBC proportions in both the basic and fully-adjusted model, this is compositional data where the increase in one blood cell type reduces the others and all add to 1. How was this data specified to the model? Including all the cell types may introduce some multicollinearity. Did the authors use Pearson correlation or Variance inflation factor scores to determine the degree of multicollinearity? How was multicollinearity handled?

* For each of the significant CpG sites, was a diagnostic such as Cooks distance used to look for highly influential data points? Often this diagnostic is useful to find allele-specific methylation.

* In section 3.4, there is some assumed knowledge. Please explain the phrase "proportional hazard assumption"

Reviewer #6: This is a well-conducted study on blood-based epigenome-wide analyses on the prevalence and incidence of 19 common disease states. The study design, datasets, statistical methods and analyses, and presentation (tables and figures) and interpretation of the results are mostly adequate and of a good standard. However, there are still a few issues needing attention.

1) In section 2.5, it says "Cox proportional-hazards models were used to adjust incident phenotypes for age at baseline and sex (17/19 phenotypes)". However, as the outcome is incident disease rather than all-cause mortality, the death becomes a competing risk in the analysis. Have the authors considered a competing risk analysis instead?

2) The poor replication across existing studies is a concern but the authors have discussed this comprehensively in the discussion.

3) Are there any multi-morbidity issues in the study, e.g. an individul developed more than one disease in the follow-up? If so, what is the impact of this interaction between diseases on the findings, and how this has been addressed in the analyses?

[LINK]

---

## [Decision Letter · Decision Letter 2]

12 May 2023

Dear Dr. Marioni,

Thank you very much for re-submitting your manuscript "Blood-based epigenome-wide analyses of nineteen common disease states: A longitudinal, population-based linked cohort study of 18,413 Scottish individuals" (PMEDICINE-D-22-04026R2) for review by PLOS Medicine.

I have discussed the paper with my colleagues and it was also seen again by 5 reviewers. I am pleased to say that provided the remaining editorial and production issues are dealt with we are planning to accept the paper for publication in the journal.

[LINK]

We look forward to receiving the revised manuscript by May 19 2023 11:59PM.   

Sincerely,

Philippa Dodd, MBBS MRCP PhD

PLOS Medicine

plosmedicine.org

Requests from Editors:

GENERAL

Thank you for your very detailed and considered responses to previous and editor and reviewer comments which the editorial team very much appreciate. Please see below for further comments that we require you address prior to publication.

*** From the Editor-in-chief – Please discuss methodological choices and approaches in respect of overfitting and the potential to find associations due to large sample size. ***

AUTHOR SUMMARY

Thank you for including an author summary which reads very nicely but is rather long. Some points currently detailed could be combined and made more concise to improve brevity while minimizing loss of information. Please revise in mind of the below guidance.

The authors summary should consist of 2-3 succinct bullet points under each of the following headings:

• Why Was This Study Done? Authors should reflect on what was known about the topic before the research was published and why the research was needed.

• What Did the Researchers Do and Find? Authors should briefly describe the study design that was used and the study’s major findings. Do include the headline numbers from the study, such as the sample size and key findings. 

• What Do These Findings Mean? Authors should reflect on the new knowledge generated by the research and the implications for practice, research, policy, or public health. Authors should also consider how the interpretation of the study’s findings may be affected by the study limitations. In the final bullet point of ‘What Do These Findings Mean?’, please describe the main limitations of the study in non-technical language.

METHODS

Line 244-258 – during our technical checks this portion of text was highlighted as overlapping with a source identified as an author PhD thesis. We appreciate that there are only so many ways that methods can be detailed and so do not find this overly concerning but would appreciate it if the authors could consider re-wording this text.

RESULTS

Are there any additional data on the underlying cause of CKD in this population? If so, it might be interesting to see how the association may appear if stratified accordingly. We suspect those data may be unavailable but if they are it could be a worthwhile exploration.

DISCUSSION

Given the multifactorial causes of CKD might the reported associations warrant further discussion perhaps in relation to the common endpoint (fibrosis)? The same would apply to liver cirrhosis. It may also be worth noting that your definition of CKD may fail to capture some individuals who would otherwise fulfil the criteria (those with proteinuria and normal eGFR, tubular disorders etc)

SUPPLEMENTARY FIGURES

We usually advise against the use of asterisks to depict p values to improve clarity, but I think in this case the converse would apply.

STROBE Checklist – I could see a title referencing its presence within the manuscript but in my version, I couldn’t find the checklist attached, please include.

SOCIAL MEDIA

If not already done so, to help us extend the reach of your research, please detail any Twitter handles you wish to be included when we tweet this paper (including your own, your coauthors’, your institution, funder, or lab) in the manuscript submission form when you re-submit the manuscript.

Comments from Reviewers:

Reviewer #1: The authors have now significantly revised this manuscript in line with both my comments and those of the other reviewers. The revised manuscript is still an excellent study with interesting findings relevant both to the disease phenotypes studies and future application of EWAS in common disease. The discussion now clearly highlights the potential limitations of the study population and analytical approach.

I have no further substantive comments to make.

Reviewer #3: I am happy that the authors have addressed the comments raised in my earlier review. Furthermore, I believe they have addressed comments raised by other reviewers.

Reviewer #4: I thank the authors for their consideration of my comments. 

Reviewer #5: The reviewers have sufficiently addressed my comments.

Reviewer #6: Many thanks authors for their great effort to improve the manuscript. All my comments/concerns were comprehensively addressed. I am satisfied with the response and revision. No further issues needing attention.

[LINK]

---

## [Editor Report · Decision Letter 3]

25 May 2023

Dear Dr Marioni, 

On behalf of my colleagues and the Academic Editor, Professor John W. Holloway, I am pleased to inform you that we have agreed to publish your manuscript "Blood-based epigenome-wide analyses of nineteen common disease states: A longitudinal, population-based linked cohort study of 18,413 Scottish individuals" (PMEDICINE-D-22-04026R3) in PLOS Medicine.

Prior to publication please ensure that you update your data availability statement to indicate that your data are now available in the EWAS Catalogue (currently 'will be made available on publication).

PRESS

Thank you again for submitting to PLOS Medicine, it has been a pleasure handling your manuscript. We look forward to publishing your paper. 

Best wishes,

Pippa 

Philippa Dodd, MBBS MRCP PhD 

PLOS Medicine